# Evaluation of Trabecular Bone Microarchitecture and Bone Mineral Density in Young Women, Including Selected Hormonal Parameters

**DOI:** 10.3390/biomedicines12040758

**Published:** 2024-03-29

**Authors:** Elżbieta Sowińska-Przepiera, Mariola Krzyścin, Igor Syrenicz, Adrianna Ćwiertnia, Adrianna Orlińska, Dorota Ćwiek, Dorota Branecka-Woźniak, Aneta Cymbaluk-Płoska, Žana Bumbulienė, Anhelli Syrenicz

**Affiliations:** 1Pediatric, Adolescent Gynecology Clinic, Department of Gynecology, Endocrinology and Gynecological Oncology, Pomeranian Medical University in Szczecin, Unii Lubelskiej 1, 71-252 Szczecin, Poland; elzbieta.sowinska.przepiera@pum.edu.pl; 2Department of Endocrinology, Metabolic and Internal Diseases, Pomeranian Medical University in Szczecin, Unii Lubelskiej 1, 71-252 Szczecin, Poland; igor.syrenicz@gmail.com (I.S.); anhelli.syrenicz@pum.edu.pl (A.S.); 3Department of Reconstructive Surgery and Gynecological Oncology, Pomeranian Medical University in Szczecin, Al. Powstańców Wielkopolskich 72, 70-111 Szczecin, Poland; adrianna.cwiertnia@gmail.com (A.Ć.); adzahorowska@gmail.com (A.O.); aneta.cymbaluk@gmail.com (A.C.-P.); 4Department of Obstetrics and Pathology of Pregnancy, Pomeranian Medical University in Szczecin, 70-204 Szczecin, Poland; dorota.cwiek@pum.edu.pl; 5Department of Gynecology and Reproductive Health, Pomeranian Medical University of Szczecin, Żołnierska 48, 71-210 Szczecin, Poland; dorota.branecka@pum.edu.pl; 6Clinic of Obstetrics and Gynecology, Institute of Clinical Medicine, Faculty of Medicine, Vilnius University, LT-08661 Vilnius, Lithuania; zana.bumbuliene@yahoo.com

**Keywords:** trabecular bone microarchitecture, bone mineral density, young women, hormonal parameters, DXA, osteoporosis, estrogen

## Abstract

The absence of non-invasive methods for assessing bone material and structural changes is a significant diagnostic challenge. Dual-energy X-ray absorptiometry (DXA) bone mineral density (BMD) testing is the gold standard for osteoporosis diagnosis. BMD and the trabecular bone score (TBS) have facilitated targeted osteoporosis prevention and treatment in clinical settings. The findings from this study indicate that BMD modulation in young women is influenced by various hormones, potentially compromising the diagnostic precision of BMD for subclinical bone demineralization. A total of 205 women aged 19 to 37 underwent anthropometric measurements and hormonal tests. BMD was determined using DXA, and TBS values were computed from the lumbar spine L1–L4 segment. The multivariate analysis findings suggest that BMD might not be determined by hormones. The relationship between TBS and TSH was statistically significant in the univariate analysis, which indicates the efficacy of further studies to determine the link between TBS and specific hormones. Analyzing the strength of the correlation between TBS and hormones in the univariate analysis shows which factors are worth considering in further analyses. This makes it possible to create better techniques that will help identify young women who are at a higher risk of developing osteoporosis.

## 1. Introduction

Bone mineral density (BMD) measurements using dual-energy X-ray absorptiometry (DXA) techniques are the most widely used and considered the gold standard for diagnosing and treating osteoporosis [1,2]. However, numerous studies have shown that BMD is not an independent and reliable predictor of osteoporotic fractures. Therefore, it is not a definitive measure of the primary characteristic that determines its quality–mechanical strength. BMD assessment includes all of the compact and spongy bone, with no ability to differentiate between the two structures. This method does not provide information on the spatial structure of spongy bone (microarchitecture), which plays a key role in determining parameters such as stiffness and elasticity, affecting the likelihood of bone fractures. These limitations have prompted the search for new, non-invasive diagnostic methods focusing on the structure of spongy bone, including its microarchitecture. A marker of spongy bone architecture and bone strength, the trabecular bone score (TBS), has been introduced into clinical practice [3]. TBS is considered equivalent to, and independent of, BMD. The TBS value is directly calculated from the results of DXA imaging of the L1–L4 lumbar segment. The use of two different markers of bone strength, BMD and TBS architecture, in clinical practice has expanded our knowledge and led to better therapeutic interventions. The study confirmed the expected relationship: assessment of bone structure with TBS predicts bone fracture risk independently of BMD and with similar probability [4]. TBS in girls during puberty and while reaching peak bone mass (PBM) is not yet widely used; therefore, definitive TBS thresholds have not been established for before PBM is reached. Previous publications indicate that the cut-off for normal TBS is a value of TBS = 1.35, reached in the first year after menarche [5]. The introduction of TBS assessment in this age group is crucial, especially in women with ovarian dysfunction. For TBS measurement to become a useful tool in the treatment of osteogenesis disorders caused by fluctuations in hormone secretion or sex hormones in younger women, it is essential to determine whether TBS is modulated by these changes and whether they are consistent directionally and quantitatively with changes in BMD. Among these factors, intrinsic factors, especially the endocrine activity of the ovaries, play an important role. Hormonal disorders during puberty lead not only to delayed puberty but also to general disruption of homeostasis, including bone formation [6,7,8]. The PBM achieved during adolescence is one of the most important predictors of postnatal osteoporosis [9,10]. The greater bone mass acquired early in life provides a specific buffer, ensuring that even if lost after menopause, it remains sufficient to maintain the mechanical function of the skeleton [11]. The greatest increase in bone mineral density occurs immediately before and after menarche [12], with peak bone mass observed in women between the ages of 20 and 25 [13,14,15,16,17,18]. The function of osteoclasts and osteoblasts is regulated by circulating external factors or locally by osteocytes, with each of these cell types influenced by different endocrine factors. The most important hormonal factors identified include PTH, vitamin D, calcitonin, TSH, thyroid hormones, growth hormone, IGF-1 and glucocorticoids [19]. Emphasis is placed on the importance of other reproductive hormones related to bone biology, such as kisspeptin, GnRH, FSH, LH, prolactin, progesterone, inhibin, activin and relaxin. In addition, adrenal androgens such as DHEA, DHEAS and androstenedione, which act as precursors for peripheral conversion to more potent androgens and estrogens, also affect bone homeostasis [20,21,22]. While numerous studies suggest a role for these hormones in bone physiology, their effects appear to influence bone mineralization through gonadal pathways. During puberty and in young women, estrogens play a fundamental role in bone remodeling. They are bound to bone cells through estrogen receptors (ER-alpha and ER-beta) that are present in all types of bone cells: osteoblasts, osteoclasts and osteocytes. The accumulation of receptors in individual cells determines the tissue’s sensitivity to estrogens [23]. The protective action of estradiol on bone tissue is bidirectional through the suppression of resorption, affecting osteoclasts, and the stimulation of mineralization involving osteoblasts; these actions occur via estrogen receptors ER-α and ER-β. Studies have demonstrated that estradiol stimulates osteoblasts to synthesize and release TGF-β, IGF-I and IGF-II while inhibiting the synthesis and secretion of cytokines such as IL-1, IL-6 and TNF, which participate in the bone resorption process. Furthermore, it has been demonstrated that ER-alpha receptors on osteocytes, derived from osteoblasts, reduce sclerostin production through ER-alpha, a signaling protein regulating osteocytogenesis, by inhibiting osteocyte apoptosis. In this situation, estrogen deficiency leads to a significant increase in osteocyte apoptosis [24]. Additionally, estrogens lower the mechanostat threshold, increasing sensitivity to mechanical stimuli. One piece of evidence supporting this is the proportional increase in bone mass with growth and body mass, particularly in girls compared to boys, which is strongly dependent on physical exercise [25,26,27]. In the bone remodeling process, formation and resorption are closely interconnected; therefore, estrogen deficiency disrupts the physiological balance between these processes, resulting in bone mass loss [28]. The trabecular bone score, a cross-sectional analysis providing indirect information about bone microarchitecture, has emerged as a potential screening tool to evaluate the factors influencing the maintenance of maximum TBS.

## 2. Materials and Methods

### 2.1. Assumptions and Objectives of the Study

The provided data indicate that the trabecular bone score has not been fully investigated in adolescent girls and young women. Both our own experiences and the results of previous studies suggest that bone mineral density in women with unstable ovarian function may be influenced by a range of hormonal parameters. This influence can negatively impact the diagnostic accuracy of BMD as a measure of subclinical depletion of bone mineralization and, indirectly, fracture risk [29,30]. Based on this, we hypothesize that certain hormonal parameters exert a significant influence on TBS as a marker of bone microarchitecture in young women. Moreover, in young women with menstrual irregularities due to abnormal secretion of sex hormones by the ovaries, there are systemic disruptions in hormonal regulation. Additionally, estrogen deficiency has a direct adverse effect on bone mineral density and potentially on bone strength. The aim of the study is to analyze the correlation of BMD with selected hormones, as well as the relationship between BMD and TBS. Potential indicators that could show a statistically significant correlation with TBS are also investigated to identify parameters that could interfere with the assessment of the likelihood of osteoporosis in young women.

### 2.2. Participation in the Study

The study included 205 women aged 19 to 37 years (mean 27.08 ± 4.33), who were patients of the Gynecological Endocrinology Outpatient Clinic of the Department of Endocrinology, Metabolic Diseases and Internal Medicine at the Pomeranian Medical University in Szczecin. The study was conducted between 2015 and 2018. Patients meeting the following inclusion criteria were included: Caucasian race, first menstruation occurred in 12–13 years of age, episodes of secondary amenorrhea lasting 3–6 months in the past year, temporary psychological problems (school and/or family and/or work), patients not on permanent medication, objectively healthy. The study protocol was approved by the Local Bioethics Committee of the Pomeranian Medical University in Szczecin (Decision No. KB-0012/115/15 dated 16 November 2015). All participants in the study have provided written consent. The criteria for exclusion were endocrinopathies affecting bone mineralization (e.g., thyroid disease, diabetes mellitus, polycystic ovary syndrome, congenital adrenal hyperplasia, premature expiration of ovarian function), which were diagnosed based on history, gynecological examination and laboratory tests. Information confirmed by history: low birth weight, prematurity, eating disorders, abnormal nutrition in childhood and/or puberty, disorders of growth and weight gain, participation in competitive sports that affect bone mineralization, metabolic diseases that may be accompanied by low bone mass, long-term use of stimulants and drugs that affect bone metabolism, positive family history of osteoporosis.

### 2.3. Basic Procedures

After taking a history and performing a routine clinical examination, all patients had anthropometric measurements taken: height [cm], weight [kg] and body mass index (BMI [kg/m^2^]) calculated.

### 2.4. Laboratory Parameters

Between the second and the sixth day of the follicular phase of the menstrual cycle, fasting venous blood drawn from the basilic vein in the morning was used to measure hormonal markers. Basic procedures after taking a history and performing a routine clinical examination, all patients had their levels determined: androstenedione, DHEAS, testosterone, SHBG, 17-OHprogesterone, luteinizing hormone (LH), folliculotropic hormone (FSH), estradiol, baseline levels of prolactin (PRL), thyrotropic hormone (TSH), triiodothyronine (fT3), thyroxine (fT4), adrenocorticotropic hormone (ACTH), cortisol, renin and aldosterone, in addition, the free androgen index (FAI) was calculated. It was determined that this group of individuals had functional hypothalamic amenorrhea (FHA), a psychogenic kind of menstruation illness that is identified by exclusion [1] after considering all of the aforementioned criteria. The blood samples were collected by qualified staff of the hospital laboratory in Department of Endocrinology, Metabolic and Internal Diseases PUM in Szczecin.

### 2.5. Bone Mineral Density Assessment

Assessment of the microarchitecture of bone beads TBS values of the same lumbar vertebra were determined from the dual-energy X-ray absorptiometry (DXA) technique images using analysis software (TBS INsight, version 2.1.2.0, Medimaps, Mérignac, France). Bone mineral density testing of the L1–L4 section of the lumbar spine and the entire skeleton was performed on all study participants. DXA (GE Lunar Prodigy Advance, Madison, WI, USA; enCORE version 8.8 software) was used to calculate BMD. The findings are presented as absolute values (g/cm^2^).

### 2.6. Statistical Analysis

The Shapiro–Wilk test was used to confirm that the continuous variable distribution was normal. The statistical features of continuous variables were displayed using the following formats: arithmetic means, medians, lower and upper quartile values, standard deviations (SD) and extreme values (Min. and Max.). The values of Spearman’s rank correlation coefficient (R) were used to assess the direction and strength of the association between pairs of continuous variables. Multiple regression analysis was used to find independent predictors of bone mineral density based on parameters that demonstrated a statistically significant (*p* ≤ 0.05) or nearly statistically significant (*p* ≤ 0.1) relationship with the dependent variables (lumbar spine barrel bone score values or bone mineral density). Using multiple regression analysis, the model’s beta values and standard errors were computed. The multiple regression analysis yielded the model’s coefficients of determination (R^2^) and *p*-values, along with beta values and associated standard errors. Bone mineral density was shown to be independently predicted by parameters that had *p*-values less than 0.05. Statistica 10 (StatSoft, Tulsa, OK, USA) was used for all calculations.

## 3. Results

### 3.1. Characteristics of the Group

The study included 205 women between the ages of 19 and 37. Detailed statistical characteristics of the patients’ age and body mass index are presented in Table 1.

The study group included 105 (51.22%) normal-weight women, 9 (4.39%) underweight, 35 (17%) overweight and 56 (27.32%) obese.

The detailed statistical characteristics of the values of the barrel bone strength score in the spine and bone mineral density of the lumbar spine are shown in Table 2 and Table 3.

#### 3.1.1. Factors Affecting Bone Mineral Density and Values of the Trabecular Bone Score in the Lumbar Spine—Results of Unidimensional Analysis

The values of Spearman’s rank correlation coefficients between bone mineral density in the lumbar spine and the values of the trabecular bone score in the same segment are shown in Table 4. The values of the trabecular bone score showed significantly positive correlations with the values of BMD L1–L4 (g/cm^2^), BMD L1–L4 (%) and z-score.

Spearman’s rank correlation coefficients between lumbar spine bone mineral density (BMD L1–L4, g/cm^2^) and hormonal parameters of the study participants are shown in Table 5.

There was a significant inverse correlation between bone mineral density and SHBG levels [*p* < 0.002; R = −0.212] and significant positive correlations among L1–L4 BMD values (g/cm^2^), FAI [*p* < 0.006; R = 0.182] and estradiol levels [*p* < 0.008; R = 0.180].

Spearman’s rank correlation coefficients between lumbar spine trabecular bone score values and hormonal parameters of the study participants are shown in Table 6. There was a significant positive correlation between trabecular bone score values and TSH levels [*p* < 0.027; R = 157].

#### 3.1.2. Results of Multivariate Analysis on Factors Influencing Lumbar Spine Bone Mineral Density

The univariate analyses presented above showed that, unlike bone mineral density, trabecular bone score values are modulated only by TSH level. Therefore, in the last phase of the investigation, the hypothesis that the trabecular bone score is an independent predictor of bone mineral density in the lumbar spine was confirmed.

The age of the patients, their SHBG levels, FAI, TSH levels and estradiol were taken into account as potential predictors of lumbar spine bone mineral density (BMD L1–L4, g/cm^2^) in multivariate regression analysis, along with the girdle bone score (Table 7). Higher values of the trabecular bone score in the same segment seemed to be the sole independent predictor of higher bone mineral density in the lumbar spine among the variables studied. R^2^ = 0.201 indicates that the proposed model only accounted for 20% of the variance in the dependent variable, although being statistically significant (*p* < 0.001).

### 3.2. Summary of Results

There was a positive correlation between TBS value and BMD L1–L4, g/cm^2^. Statistically significant positive correlations occurred between BMD with estradiol and FAI, and a reversed correlation was found between BMD and SHBG. On the other hand, BMD L1–L4, g/cm^2^ was positively significantly correlated only with TSH.

## 4. Discussion

Some studies show that TBS can predict fracture risk in adults with low BMD or poor bone quality [31]. Other studies in adult populations have evaluated the usability of a combined TBS and BMD assessment to improve fracture risk prediction. Moreover, there was a predictive association of TBS with spinal fractures in women without osteoporosis, even though BMD did not detect osteoporotic fractures. The authors of this study take a cautious stance, believing that larger prospective studies on this subject are needed to confirm the currently emerging outcomes in this area [32,33]. Therefore, the study of factors affecting TBS in the young female population is an attempt to search for a new independent, attractive and non-invasive marker for assessing bone quality.

Estrogen deficiency is a well-documented risk factor for bone mass loss [34,35,36]. Skeletal demineralization may also occur because SHBG preferentially binds androgens and fails to convert them further to estrogen [37,38,39]. Accepting such a mechanism of action would explain why BMD in the study group correlates inversely with SHBG and increases with FAI values.

The SWAN TBS (The Study of Women’s Health Across the Nation) study involved 705 Caucasian, Black and Japanese women in the early peri-menopausal period who had experienced menstrual cessation. TBS loss in Caucasian women began 1.5 years before menopause and decreased by 1.16% per year (*p* < 0.0001) to 0.89% after 2 years (*p* < 0.0001). The total decrease in TBS after 5 years in Caucasian women was 6.3% (*p*< 0.0001), while the total loss of TBS in black women was lower, 4.90% (*p* = 0.0008). TBS scores in Japanese women did not differ compared to those of white women. The authors concluded that the decrease in TBS already occurs during the menopausal transition, which is important for preventive treatment relevant to skeletal mineralization [40]. Lower TBS values are also observed in women in their 20s and 40s with premature ovarian insufficiency function [41].

Bone strength in endocrine disorders can be an important diagnostic indicator since BMD values do not always reflect bone microarchitecture disorders. Understanding bone microarchitecture significantly increases the body of knowledge regarding the pathophysiology of primary and secondary osteoporosis. Nevertheless, bone quality cannot be comprehensively characterized by only one parameter. Presumably, the integration of the combined use of densitometric parameters of BMD, TBS and clinical risk factors will prove more useful in the diagnosis of skeletal health.

The trabecular bone score values are not influenced by hormonal parameters other than TSH levels, in contrast to bone mineral density, according to the study’s stated univariate analysis. Thyroid hormones play a key role in regulating bone metabolism and development. They contribute to maintaining bone structure and strength and achieving peak bone mass [42]. Thyroid hormones have been shown to affect the time course of the remodeling cycle and the interplay between bone formation and resorption. During adolescence and bone growth, on the length and its mineralization, thyroid hormones have an anabolic effect, while in adults, they exert mainly catabolic effects [43]. The associations studied between lumbar spine trabecular bone score values and hormonal parameters of the study participants were non-significant, while a relevant positive correlation was found between trabecular bone score values and TSH levels. Most studies on the effects of the TSH–T3–T4 system on bone mineralization indicate that TSH inhibits the differentiation and function of osteoclasts; in addition, TSH has also been shown to have an inhibitory effect on chondrocytes and osteoblasts. Some publications describe the inhibitory effects of TSH on osteoclastogenesis and bone resorption through direct action on the TSH receptor of osteoclasts. Therefore, it might have a protective role on bone [44]. Other authors have shown that TSH may have a qualitative protective effect on bone mineralization in a thyroid hormone-independent manner through the PTH–calcium–sclerostin regulatory mechanism [45]. Sclerostin is a recently identified glycoprotein synthesized by osteocytes and the product of the SOST gene. This protein is already a documented inhibitor of osteoblast proliferation and differentiation, which can lead to bone demineralization [46]. Studies have suggested that one of the important factors that lead to an increase in sclerostin secretion is fluctuations in estrogen concentrations and deficiency, as well as a reduction in bone loading (mechanical stress), leading to a significant loss of bone mass [47,48]. It has also been noted that when thyroid hormone concentrations fluctuate (e.g., thyroid hormone excess), sclerostin levels are high, and after treatment of thyrotoxicosis, sclerostin concentrations decrease [49,50]. Studies by other authors show that excess PTH resulted in a reduction in sclerostin levels, so it is reasonable to assume that a reduction in its concentration contributes to an increase in bone mineral density [51,52]. Thyrotropin may have an indirect positive effect on bone mineralization through a mechanism involving triiodothyronine (T3), as triiodothyronine concentrations have been shown to be associated with better bone quality characteristics [53]. The regulatory mechanisms described above in this study are poorly documented, as there was no significant correlation among fT3, TBS or BMD, nor did TSH prove to be an independent predictor of bone architecture in multivariate analysis. It cannot be ruled out that the positive correlation between TSH and TBS is mediated by leptin, which was not measured, as one previous study showed that adipokine synthesized in subcutaneous adipose tissue, especially in the gynoid area, positively correlates with TSH values and inversely with fT4 in both adults and children [54,55,56,57].

The few studies, mainly on the elderly female population, suggest that TBS value, as an independent parameter, may be useful in assessing bone mineralization and predicting fractures in the course of osteoporosis secondary to endocrine causes. Other interesting studies involve patients with acromegaly, who are at risk for fractures regardless of BMD values. Lower TBS is observed in these patients compared to healthy persons [58]. Previous findings have suggested that excess GH may positively affect cortical bone and negatively affect trabecular bone. Additionally, the treatment of patients with acromegaly had different effects on TBS and BMD. Over the course of treatment, TBS values decreased significantly while BMD increased, so in acromegaly, assessing bone quality using BMD may be misleading [59]. Other studies have shown that TBS might be a predictor of fractures independent of BMD in patients with primary hyperparathyroidism (PHPT) [60]. Patients with fractures and PHPT had lower TBS than patients without fractures. Furthermore, the TBS value improved significantly after parathyroidectomy compared to conservative treatment of PHPT [61,62,63,64]. TBS testing was a more useful parameter in assessing bone quality in patients with excess cortisol. Hypercortisolemia has been shown to impair bone mineralization; however, only a slight decrease in BMD was noted, while the decrease in TBS was more pronounced [65,66]. An interesting study in this regard concerns TBS as a diagnostic factor in early, subclinical hypercorticolemia. It was also revealed that high evening cortisol values were associated with low TBS and increased fracture rates in healthy postmenopausal women. This evidence suggests the usefulness of TBS in assessing bone health in a spectrum of cortisol disorders [67,68]. Studies of TBS in patients with hypogonadism are included in a few studies. One study focused on patients with Klinefelter syndrome and hypogonadism, in which there was no difference in TBS values compared to the control group, while BMD was reduced. Treating patients with testosterone preparation and evaluating them in a three-year sequence did not improve TBS values, while the BMD parameter increased [69]. A similar mechanism occurred among the group of women examined in this study, where BMD correlated positively with the FAI-free androgen index.

This study also has its limitations. The primary limitation of this study was its retrospective nature, as a result of which, it was not possible to exclude the potential influence on BMD and TBS values of additional biochemical parameters (such as leptin levels) and a number of potential determinants of bone quality (such as diet, physical activity). For this reason, it was not possible to form a control group. Nevertheless, due to the appropriate choice of statistical methodology (correlation and regression analysis rather than intergroup comparisons) and the large sample size, the findings presented here are reliable, and the assumptions are supported by their considerable consistency with published evidence.

## 5. Conclusions

In summary, the multivariate analysis findings suggest that BMD might not be determined by hormone factors. Among the variables analyzed, the only independent predictor of higher bone mineral density in the lumbar spine was a higher trabecular bone score in the same lumbar section. The relationship between TBS and TSH was statistically significant in the univariate analysis, which indicates the efficacy of further studies to determine the link between TBS and specific hormones. The study’s findings are used as the basis for identifying potential BMD-influencing factors. Analyzing the strength of the correlation between TBS and particular hormones in the univariate analysis shows which factors are worth considering in further analyses. This makes it possible to create more precise techniques that will help identify young women who are at a higher risk of developing osteoporosis.

## Figures and Tables

**Table 1 biomedicines-12-00758-t001:** Statistical characteristics of age and body mass index in female study participants. *n*—number of participants; SD—standard deviation; Q_1_—first quartile; Q_3_—third quartile.

Variable	*n*	Mean	SD	Median	Q_1_	Q_3_	Min.	Maks.
Age (years)	205	27.08	4.33	27.00	24.00	30.00	19.00	37.00
BMI (kg/m^2^)	205	25.60	5.82	23.80	20.90	30.00	16.22	45.50

**Table 2 biomedicines-12-00758-t002:** Statistical characteristics of bone mineral density and values of the barrel bone strength score at the lumbar spine in the women studied. *n*—number of participants; SD—standard deviation; Q_1_—first quartile; Q_3_—third quartile.

Variable *n* = 205	Mean	SD	Median	Q_1_	Q_3_	Min.	Maks.
BMD L1–L4 (g/cm^2^)	1.23	0.13	1.24	1.15	1.32	0.83	1.58
BMD L1–L4 (%)	102.51	9.95	103.00	96.00	109.00	71.00	125.00
BMD z L1–L4 score	0.23	0.98	0.30	−0.40	1.00	−2.80	2.50
TBS L1–L4	1.38	0.09	1.38	1.32	1.43	1.18	1.70

**Table 3 biomedicines-12-00758-t003:** The statistical characteristics of hormonal parameters for study participants. *n*—number of participants; SD—standard deviation; Q_1_—first quartile; Q_3_—third quartile.

Variable *n* = 205	Mean	SD	Median	Q_1_	Q_3_	Min.	Maks.
Androstenedione	3.94	1.91	3.60	2.85	4.78	1.00	17.10
DHEA	260.07	125.48	254.00	178.00	325.00	10.68	781.10
Testosterone	0.50	0.22	0.48	0.36	0.62	0.06	1.24
SHBG	56.11	45.98	45.50	28.34	70.70	6.80	399.60
FAI	5.29	5.18	3.94	2.07	6.46	0.00	39.89
17-OHP	1.17	0.65	1.10	0.76	1.47	0.31	7.51
LH	9.02	7.66	6.93	4.74	11.15	0.10	73.82
FSH	6.13	5.75	5.57	4.59	6.71	0.12	78.31
Estradiol	69.42	81.71	45.34	32.61	70.49	5.00	329.40
PRL 0′	20.11	24.06	16.65	11.20	22.40	1.46	336.70
PRL 60′	166.18	73.81	157.80	121.70	189.80	6.60	551.00
TSH	2.40	3.50	1.81	1.31	2.83	0.02	48.03
fT3	3.05	0.35	2.95	2.80	3.34	2.40	3.97
fT4	1.44	1.80	1.23	1.12	1.33	0.85	20.57
Cortisol	16.63	6.26	15.75	12.40	20.19	7.30	50.40
ACTH	34.41	47.84	26.49	19.42	38.00	1.00	481.00

**Table 4 biomedicines-12-00758-t004:** Spearman rank correlation coefficients (R) between trabecular bone strength indices in the lumbar spine and bone mineral density in the same region.

Variable *n* = 205	R	*p*
BMD L1–L4 (g/cm^2^)	0.334	<0.001
BMD L1–L4 (%)	0.270	<0.001
z-score	0.263	<0.001

Statistical significance: *p* ≤ 0.05.

**Table 5 biomedicines-12-00758-t005:** Spearman rank correlation coefficients (R) between bone mineral density in the lumbar spine (BMD L1–L4, g/cm^2^) and hormonal parameters in study participants.

Variable *n* = 205	R	*p*
Androstenedione	−0.038	0.585
DHEA	0.046	0.503
Testosterone	0.072	0.295
SHBG	−0.212	0.002
FAI	0.192	0.006
17-hydroxyprogesterone	0.047	0.492
LH	−0.048	0.487
FSH	−0.093	0.175
Estradiol	0.180	0.008
PRL 0′	0.076	0.267
PRL 60′	0.054	0.429
TSH	−0.065	0.359
fT3	−0.014	0.884
fT4	0.092	0.205
Cortisol	0.099	0.263
ACTH	0.049	0.592

Statistical significance: *p* ≤ 0.05.

**Table 6 biomedicines-12-00758-t006:** Spearman rank correlation coefficients (R) between trabecular bone score (TBS) in the lumbar spine and hormonal parameters in study participants.

Variable *n* = 205	R	*p*
Androstenedione	−0.066	0.341
DHEA	−0.046	0.509
Testosterone	−0.032	0.648
SHBG	−0.072	0.298
FAI	0.044	0.534
17-hydroxyprogesterone	−0.020	0.770
LH	−0.029	0.672
FSH	0.016	0.823
Estradiol	−0.099	0.154
PRL 0′	−0.043	0.541
PRL 60′	−0.059	0.398
TSH	0.157	0.027
fT3	−0.147	0.114
fT4	0.054	0.458
Cortisol	0.084	0.351
ACTH	0.012	0.893

Statistical significance: *p* ≤ 0.05.

**Table 7 biomedicines-12-00758-t007:** Factors affecting lumbar spine bone mineral density (BMD L1–L4, g/cm^2^), results of multiple regression analysis.

Variable	*n* = 205 Beta	Standard Error of Beta	*p*
TBS L1–L4	0.290	0.076	<0.001
Age (years)	0.132	0.071	0.066
SHBG	−0.032	0.080	0.693
FAI	−0.044	0.084	0.605
Estradiol	0.092	0.070	0.190
TSH	−0.103	0.071	0.148
BMI (kg/m^2^)	0.212	0.140	0.130

Statistical significance: *p* ≤ 0.05.

## Data Availability

The data presented in this study are available on request from the corresponding author, M.K., upon reasonable request.

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
