# Peer review of "Evaluation of Trabecular Bone Microarchitecture and Bone Mineral Density in Young Women, Including Selected Hormonal Parameters"

_biomedicines, 2024, doi:10.3390/biomedicines12040758_

Round 1
Reviewer 1 Report
Comments and Suggestions for Authors
11. The conclusion of this study is not supported by their results. The authors stated, "while BMD correlated positively with estradiol FAI and inversely correlated with SHBG." However, this statement was not supported by the multivariable analysis results when adjusted for TBS and age.
22. BMI, which was related to both BMD and hormones, should be adjusted in the multivariable analysis.
33. The authors stated that "TBS is a more reliable determinant of bone quality because, unlike bone mineral density (BMD), this value is independent of the interfering effect of hormonal parameters." However, you stated that TBS was influenced by THS; therefore, this statement was not correct. Furthermore, the multivariable analysis results indicate that BMD was indeed independent of hormonal parameters.
Author Response
Dear Reviewer,
Thank you very much for your comments. They certainly improved the quality of our article. We have made every effort to answer them. Below is the answer to your comment.
- The conclusion of this study is not supported by their results. The authors stated, "while BMD correlated positively with estradiol FAI and inversely correlated with SHBG." However, this statement was not supported by the multivariable analysis results when adjusted for TBS and age.
Response: According to your valuable comment, we changed the conclusion of this study.
“In summary, the multivariate analysis's findings suggest that BMD might not be determined by hormone factors. Among the variables analyzed, the only independent predictor of higher bone mineral density in the lumbar spine was a higher trabecular bone score in the same lumbar section. The relationship between TBS and TSH was shown statistically significantly in the univariate analysis, which indicates the efficacy of further studies, looking for the link between TBS and specific hormones. The studies' findings are used as the basis for identifying potential BMD-influencing factors. Analyzing the strength of the correlation between TBS and particular hormones in the univariate analysis shows which factors are worth considering in further analysis. This makes it possible to create more precise techniques that will help identify young women who are at a higher risk of developing osteoporosis.”
- BMI, which was related to both BMD and hormones, should be adjusted in the multivariable analysis.
Response: Indeed, this information was lacking. We added information on BMI as a predictor of bone mineral density in the lumbar spine checked in the multivariate analysis according to your suggestion.
- The authors stated that "TBS is a more reliable determinant of bone quality because, unlike bone mineral density (BMD), this value is independent of the interfering effect of hormonal parameters." However, you stated that TBS was influenced by THS; therefore, this statement was not correct. Furthermore, the multivariable analysis results indicate that BMD was indeed independent of hormonal parameters.
Response: We appreciate your insightful suggestion. Therefore, we have modified the sentence suggesting that TSH affects the TBS as below.
“The relationship between TBS and TSH was shown statistically significantly in the univariate analysis, which indicates the efficacy of further studies, looking for the link between TBS and specific hormones. In this study, we focused more on the influence of selected hormonal parameters on BMD values. Since a multivariate analysis is required to verify the impact of TSH on TBS more studies are required. We also added a new statement.”
I hope that you will find the revisions appropriate and allow the manuscript to be published.
Kind regards,
Adrianna Ćwiertnia
Reviewer 2 Report
Comments and Suggestions for Authors
The authors suggest utilize TBS index in addition or instead of BMI in clinical settings for young women.
Comments
1. The authors should unify the abbreviation for TBS: either trabecular bone score or trabecular bone strength index.
2. Sentence on Line 32 contradicts with that on Line 33-34. This should be corrected.
3. Lines 52, 65 etc: All abbreviations should be disclosed on first use.
4. References are required after every statement: Lines 60,65, 106,117, 119,
5. Line 52: The definition and description of TBS should be provided here.
6. The language of the paper should be improved as some sentences are not clear. For example, lines 63-65, 351-356, 460-461.
7. Lines 141-144: The aim of the study is not clear. It should be clarified.
8. Tables and Figures: The data should not be repeated in Tables and Figures. The authors should choose one of the presentation types. As the importance of non-significant data is negligible the authors should mention these results in the text and present significant correlations on graphs. Or they should leave only Tables and withdraw the Graphs. In addition, the number of patients should be indicated under each graph and in the caption of each table.
9. Discussion should not repeat Results section (Lines 334-345)
10. All the repeated sentences should be removed. Example: Lines 380-381 versus 393-394
11. Lines 393-394: It is not clear why inhibition of bone metabolism/turnover is protective? Bone remodeling is an important requirement for bone health. This should be clarified.
12. Lines 431-438: It is not clear why the authors failed to find association between TSH and cortisol if even during subclinical hypercorticolemia this association was observed. This should be clarified.
Comments on the Quality of English Language1. The language of the paper should be improved as some sentences are not clear. For example, lines 63-65, 351-356, 460-461.
2. Lines 141-144: The aim of the study is not clear. It should be clarified.
1. All the repeated sentences should be removed. Example: Lines 380-381 versus 393-394
Author Response
Dear Reviewer,
Thank you very much for your comments. They certainly improved the quality of our article. We have made every effort to answer them. Below there is the answer to your comment.
- The authors should unify the abbreviation for TBS: either trabecular bone score or trabecular bone strength index.
Response: Indeed, this information was lacking. We modified all trabecular bone strength index to trabecular bone score.
- Sentence on Line 32 contradicts with that on Line 33-34. This should be corrected.
Response: We analyzed all the aims and conclusions of this study and changed inappropriate sentences.
The multivariate analysis's findings suggest that BMD might not be determined by hormones. The relationship between TBS and TSH was shown statistically significantly in the univariate analysis, which indicates the efficacy of further studies, looking for the link between TBS and specific hormones.
- Lines 52, 65 etc: All abbreviations should be disclosed on first use.
Response: We have expanded the abbreviations used for the first time according to your suggestion.
- References are required after every statement: Lines 60,65, 106,117, 119,
Response: Indeed, we concur, much appreciated for your astute observation. We have completed the references. At the suggestion of another reviewer, we shortened the introduction.
- Line 52: The definition and description of TBS should be provided here.
Response: We have added the description of TBS.
Now line 58.
- The language of the paper should be improved as some sentences are not clear. For example, lines 63-65, 351-356, 460-461.
Response: Thank you for your valuable comments. We corrected the indicated sentences and attempted to improve the language quality.
We removed lines 63-65, due to the fact that another reviewer suggested that we should shorten the introduction.
We changed lines 351-356: More prospective researches has to be done in this field to verify these early results. Now lines 302-304.
We also changed lines 460-461 to: The results of the study might help in developing preventive measures to identify young women who are at risk for osteoporosis. Now lines 421-423.
- Lines 141-144: The aim of the study is not clear. It should be clarified.
Response: According to your suggestion we have changed the aim of the study.
The aim of the study is to analyze the correlation of BMD with selected hormones, as well as the relationship between BMD and TBS. Potential indicators that could show a statistically significant correlation with TBS were also investigated to identify parameters that could interfere with the assessment of the likelihood of osteoporosis in young women.
- Tables and Figures: The data should not be repeated in Tables and Figures. The authors should choose one of the presentation types. As the importance of non-significant data is negligible the authors should mention these results in the text and present significant correlations on graphs. Or they should leave only Tables and withdraw the Graphs. In addition, the number of patients should be indicated under each graph and in the caption of each table.
Response: Your observation is on point, thank you for your valuable comment. We have removed all Figures and left the Tables. Relevant information that was previously included in the table has been presented in the text. We also added the number of patients in the caption of each table.
- Discussion should not repeat Results section (Lines 334-345)
Response: Valid point. We have removed lines 334-345 from the discussion section.
- All the repeated sentences should be removed. Example: Lines 380-381 versus 393-394
Response: The article has been analyzed and repetitions have been removed.
- Lines 393-394: It is not clear why inhibition of bone metabolism/turnover is protective? Bone remodeling is an important requirement for bone health. This should be clarified.
Response: We agree with your wise suggestion that bone remodeling is important for bone health. Therefore we added a sentence explaining the effect of TSH on bone cells.
“There are publications that describe the inhibitory effects of TSH on osteoclastogenesis and bone resorption through direct action on the TSH receptor of osteoclasts. Therefore, it might have the protective role of bone”
- Lines 431-438: It is not clear why the authors failed to find association between TSH and cortisol if even during subclinical hypercorticolemia this association was observed. This should be clarified.
Response: In consideration of a potential typographical error (TSH was intended to be TBS). We focused on the content indicated in lines 431-438. In this part of the discussion, we wrote about the association of hypercortisolemia with TBS, which in the cited papers was verified in people with Cushing's syndrome or disease. In this study, it was checked whether there was a statistically significant or close to statistically significant association between the compared groups of the predictor and TBS. Moreover, the psychogenic type of menstrual disorder functional hypothalamic amenorrhea was diagnosed in the analyzed group using exclusion criteria, including endocrinopathy affecting bone mineralization.
Comments on the Quality of English Language
- The language of the paper should be improved as some sentences are not clear. For example, lines 63-65, 351-356, 460-461.
Response: Indeed. We have verified the article and improved the language.
- Lines 141-144: The aim of the study is not clear. It should be clarified.
Response: We have changed the aim of the study.
The aim of the study is to analyze the correlation of BMD with selected hormones, as well as the relationship between BMD and TBS. Potential indicators that could show a statistically significant correlation with TBS were also investigated to identify parameters that could interfere with the assessment of the likelihood of osteoporosis in young women.
- All the repeated sentences should be removed. Example: Lines 380-381 versus 393-394
Response: The discussion section on results (374-387, so including lines 380-381) has been removed to avoid repetition from the subheading "Results."
I hope that you will find the revisions appropriate and allow the manuscript to be published.
Kind regards,
Adrianna Ćwiertnia
Reviewer 3 Report
Comments and Suggestions for Authors
Dear Authors,
many compliments for Your paper. Your effort to deepen the osteoporosis diagnosis is useful for the scientific community involved in this field.
The article seems well structured and organized in all its sections. The statistical analyses is well carried out. You also clearly highlighted its limitations.
I suggest just some minor corrections.
Check that in the abstract all the acronymous are explained when they appear for the first time in the text.
Then, the introduction seems too long and repetitive, you should make this section shorter.
Best regards
Author Response
Dear Reviewer,
Thank you very much for your comments. They certainly improved the quality of our article. We have made every effort to answer them. Below is the answer to your comment.
- Check that in the abstract all the acronymous are explained when they appear for the first time in the text.
Response: We have added all the acronymous’ explanation. - Then, the introduction seems too long and repetitive, you should make this section shorter.
Response: We have shortened the introduction and removed repetitions.
I hope that you will find the revisions appropriate and allow the manuscript to be published.
Kind regards,
Adrianna Ćwiertnia
Round 2
Reviewer 1 Report
Comments and Suggestions for Authors
No more comments
